# Antiviral activity of intracellular nanobodies targeting the influenza virus RNA-polymerase core

**Mélissa Bessonne[1]☯, Jessica Morel[1]☯, Quentin Nevers[1], Bruno Da Costa[1], Allison Ballandras-Colas[2], Florian Chenavier[2], Magali Grange[3], Alain Roussel[3], Thibaut Crépin[2], Bernard Delmas[1]***

**1** Unité de Virologie et Immunologie moléculaires, INRAE, Université Paris-Saclay, Jouy-en-Josas, France, **2** Institut de biologie structurale, CNRS, Université de Grenoble, Grenoble, France, **3** Laboratoire d'Ingénierie des Systèmes Macromoléculaires (LISM), CNRS, Université d'Aix-Marseille, Marseille, France

☯ These authors contributed equally to this work.
* bernard.delmas@inrae.fr

**Data Availability Statement:** Data used for this submission are accessible on the link: https://www.ebi.ac.uk/biostudies/studies/S-BSST1359.

## Abstract

Influenza viruses transcribe and replicate their genome in the nucleus of the infected cells, two functions that are supported by the viral RNA-dependent RNA-polymerase (FluPol). FluPol displays structural flexibility related to distinct functional states, from an inactive form to conformations competent for replication and transcription. FluPol machinery is constituted by a structurally-invariant core comprising the PB1 subunit stabilized with PA and PB2 domains, whereas the PA endonuclease and PB2 C-domains can pack in different configurations around the core. To get insights into the functioning of FluPol, we selected single-domain nanobodies (VHHs) specific of the influenza A FluPol core. When expressed intracellularly, some of them exhibited inhibitory activity on type A FluPol, but not on the type B one. The most potent VHH (VHH16) binds PA and the PA-PB1 dimer with an affinity below the nanomolar range. Ectopic intracellular expression of VHH16 in virus permissive cells blocks multiplication of different influenza A subtypes, even when induced at late times post-infection. VHH16 was found to interfere with the transport of the PA-PB1 dimer to the nucleus, without affecting its handling by the importin β RanBP5 and subsequent steps in FluPol assembly. Using FluPol mutants selected after passaging in VHH16-expressing cells, we identified the VHH16 binding site at the interface formed by PA residues with the N-terminus of PB1, overlapping or close to binding sites of two host proteins, ANP32A and RNA-polymerase II RPB1 subunit which are critical for virus replication and transcription, respectively. These data suggest that the VHH16 neutralization is likely due to several activities, altering the import of the PA-PB1 dimer into the nucleus as well as inhibiting specifically virus transcription and replication. Thus, the VHH16 binding site represents a new Achilles' heel for FluPol and as such, a potential target for antiviral development.

**Funding:** M.B. acknowledges fellowships of the DIM1Health and of the Animal Health Division of INRAE, J.M. acknowledges fellowships of the ANR program and the Animal Health Division of INRAE. B.D. acknowledges support of the ANR-17-CE18-0006-01 program. The funders had no role in study design, data collection and analysis, decision to publish, or preparation of the manuscript.

**Competing interests:** The authors have declared that no competing interests exist.

## Author summary

The influenza virus RNA-polymerase (FluPol) ensures genome transcription and replication in the nucleus of the infected cells. To select ligands able to block FluPol activities, we screened a phages library encoding nanobodies and resulting from the immunization of a llama with FluPol subunits. When expressed intracellularly, one of the nanobodies displays highly efficient FluPol blocking and virus neutralizing activity. This nanobody recognizes the PA subunit and the PA-PB1 dimer with high affinity. The VHH binds an epitope overlapping or close to the binding sites of ANP32A and RNA-polymerase II, two key cell proteins involved in FluPol replication and transcription activities, respectively. Furthermore, the VHH was found to interfere with the transport of the PA-PB1 dimer into the nucleus. Thus, the high inhibitory activity of the VHH may result from diverse properties, altering FluPol subunits trafficking inside the cell and FluPol transcription and replication. Targeting the VHH binding site may constitute a powerful strategy to develop new antivirals.

## Introduction

Seasonal human influenza viruses epidemics and pandemics in a recurrent mode cause significant morbidity and represent a main public health burden every year in the world. Furthermore, animal influenza viruses infecting pigs and avian species can compromise meat industries, as exemplified by the 50 million birds culled in the affected establishments with circulation of highly pathogenic avian influenza viruses in poultry, captive and wild birds in the 2021–2022 epidemic season in Europe [1].

Influenza A viruses genome is made of eight RNA segments of negative polarity packaged in viral ribonucleoprotein (vRNP) complexes (reviewed in [2]). Each of these vRNPs is composed of a large number of copies of the nucleoprotein (N) associated to genomic RNAs, with a single virus RNA-dependent RNA-polymerase (FluPol) bound to the 5'- and 3'-ends of the RNA segment. The vRNPs constitute the structural and functional units for genome replication and transcription which occur in the nucleus of the infected cell. The three largest segments encode the FluPol subunits: the two basic proteins PB1 and PB2 and the acidic subunit PA (reviewed in [3]). The polymerase subunits, which are produced in the cytoplasm, are then imported into the nucleus and assembled into a functional trimer [4]. PA and PB1 form a dimer in the cytoplasm, which is imported into the nucleus separately from PB2 [5], [6], [7, 8]. Once in the nucleus, the PB1-PA dimer associates with PB2 to form the heterotrimeric polymerase. The nucleotide polymerization activity is common to both replication and transcription, with an additional cap-snatching function being employed during transcription to steal short 5'-capped RNA primers from host mRNAs [9].

The PB1 subunit functions as the polymerase catalytic subunit. It binds to the promoter sequences of the viral and complementary RNAs, and catalyzes RNA chain elongation [10], [11]. The PB2 subunit is responsible for recognition and binding of the cap structure of host mRNAs [12], [13]. The PA subunit is divided into two main domains structurally well defined, the endonuclease domain (amino acids 1 to 197) and a large C-terminal domain (amino acids 257 to 716) that defines, with the PB1 subunit, the core of FluPol with the 237 N-ter residues of PB2. The PA endonuclease and the PB2 cap-binding domains are flexibly-linked domains to the core and they act synergistically to promote cap-snatching-dependent transcription [14].

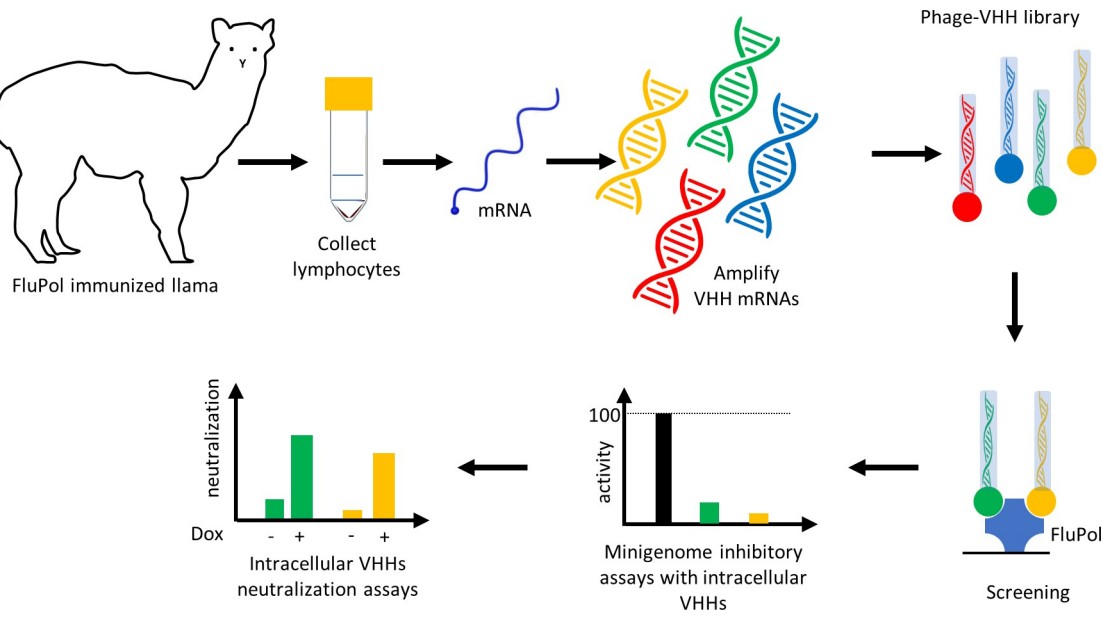

**Fig 1. Selection mode of intracellular neutralizing FluPol-specific nanobodies/VHHs.**

Nanobodies (also known as VHHs), which are variable domains of heavy chain-only camelid antibodies of about 15kDa, have proved to be versatile molecular binders for their use as research tools in structural and cell biology [15], [16]. Their ability to bind their cognate ligand is not dependent of post-translational modifications such as disulfide bonds and glycosylation [17]. These properties allow the VHHs to be expressed in the cytosol of eukaryotic cells with preservation of their binding activity properties [18]. The intracellular expression of a VHH specific of a viral protein involved in any key function in the virus cycle may perturbate the target protein functioning and block virus multiplication (reviewed in [19]). Understanding the molecular bases involved in the inhibition of the virus replication may provide new perspectives in the design of antivirals targeting influenza viruses.

In this work, we generated nanobodies specific to the structurally rigid core domain of FluPol and identified a VHH able to efficiently block FluPol activity and virus multiplication (see **Fig 1** for the rationale of the study). This VHH recognizes an interface constituted of PA residues with the PB1 N-terminus close to binding sites of two host proteins, ANP32A and RNA--Polymerase II RPB1, both critical for virus replication and transcription, respectively.

## Results

### Generation of VHHs specific to the influenza virus RNA-polymerase

To generate VHHs specific to the influenza virus RNA-polymerase, a llama was immunized with the core of an H3N2 FluPol constituted by the full-length PB1 subunit with the two-third C-terminal moieties of PA (197–716) and PB2 amino acids (1–116) [20] (**Fig 2A**). After five rounds of immunization, peripheral mononuclear cells were collected. VHH-encoding sequences were amplified from purified RNA and cloned into an M13 phagemid vector to create a specific nanobody phage library. Screening of nanobodies was performed by phage display as previously published [15], [21]. Three rounds of panning allowed the selection of six FluPol specific binders (**Fig 2B**).

A

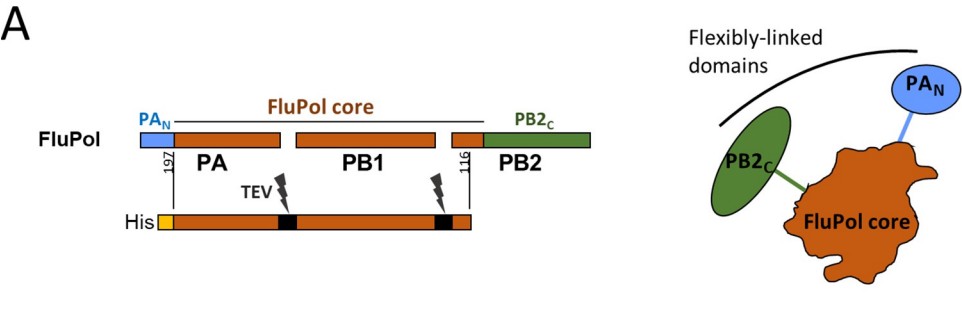

B

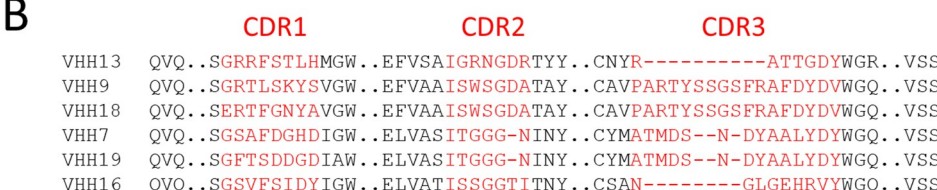

**Fig 2. FluPol polypeptides and FluPol-specific VHH sequences. A.** Schematic representation of FluPol and its structural domains, the core (colored in brown) constituted by PB1 and domains of the PA and PB2 subunits and flexibly-linked domains, $PA_N$ and $PB2_C$ (in blue and green colors) that display different packing arrangements onto the core. To select core-specific VHHs, the FluPol core was expressed as a fusion protein with a His-tag and linkers (black colors) cleavable by the TEV protease expressed in frame [20]. The core was produced and purified for alpaca immunization to generate a VHH library screening. **B.** ClustalW-based amino acids alignment of the anti-FluPol core VHHs with their CDR1, CDR2 and CDR3 domains represented in red characters.

## Intracellular VHHs potency on FluPol activity

Using a minireplicon assay, we assessed the effects of intracellular expression of VHH cDNAs on the transcription/replication activity of the viral polymerase. The pPol1-WSN-NA-firefly luciferase plasmid produces a modified influenza NA viral RNA (vRNA) in which the NA-coding sequences are replaced by the firefly luciferase gene. Upon co-transfection of HEK-293T cells of the pPOL1-WSN-NA-firefly luciferase plasmid with plasmids allowing expression of PA, PB1, PB2 and NP with one VHH, the luciferase enzymatic activity measured in cell extracts reflects the overall transcription and replication activities of the transiently expressed viral polymerase. This assay revealed that two VHHs, VHH16 and VHH18 have strong potency in a cellular context, VHH16 being able to block up to 95% of FluPol activities of an H3N2 virus and the WSN (H1N1) virus (**Figs 3A and S1**). Expression of VHH open reading frames was validated by an indirect immune-fluorescence assay (**S2 Fig**). We noted that VHH13 was less easily detected than others, suggesting a lower intracellular stability. **Fig 3B** shows that VHH16 inhibitory activity was influenza type-specific, as it did not block influenza type B FluPol. We further established that FluPol inhibition by VHH16 was dose-dependent and that a 1/25 ratio between the plasmid expressing VHH16 and the ones expressing each FluPol subunit blocks 50% of the FluPol activity (**Fig 3C**).

## VHH16 inhibits influenza A virus multiplication *in cellulo*

Having shown the ability of two VHHs to inhibit FluPol activity in cells, we wondered whether they would be able to inhibit viral multiplication. For this purpose, we first used a recombinant bioluminescent reporter virus, with the nanoluciferase gene inserted in frame into the PA genomic segment, to quantify viral multiplication [22]. We first assessed the inhibitory effect of VHHs in a transient transfection assay in HEK-293T cells, in which VHH-encoding plasmids were transfected 24 hrs before infection (**S3 Fig**). While VHH18, 7 and 9 were not able to

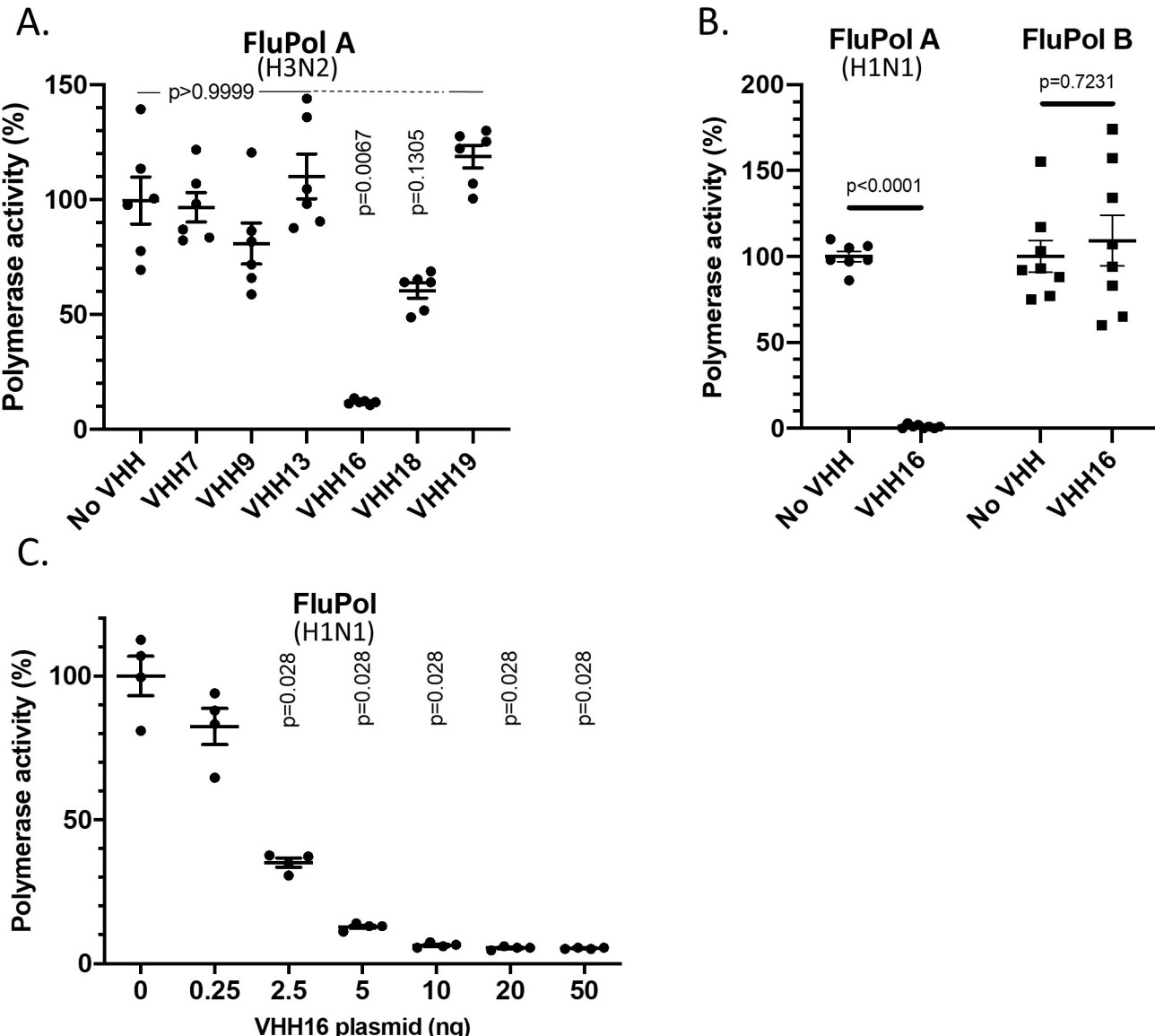

**Fig 3. Effect of VHH expression on the FluPol replication and/or transcription activities.** (**A**) Effect of VHHs on FluPol activity using a luciferase-reporter minireplicon assay. Plasmids expressing NP, PA, PB1, PB2 of a H3N2 strain were co-transfected in HEK 293T cells together with the WSN-NA-firefly-luciferase reporter plasmid and with a plasmid encoding a VHH or with an empty plasmid (indicated as No VHH). A plasmid encoding the nano-luciferase was co-transfected to control DNA uptake and normalize minireplicon activity. Luciferase activities were measured in cell lysates 48 hours post-transfection. (**B**) Same procedure than in (**A**), except that plasmids encoding the replicative complex of the WSN H1N1 strain and an influenza B virus (strain B/Memphis/13/2003) with its replicon were included in the assay. In (**A and B**), data are mean ± s.e.m. n = 2 independent transfections with n = 3 or 4 technical replicates. Kruskal-Wallis test was used to compare luminescence in the presence or absence of nanobodies. $p < 0.05$ is considered significant. In (**C**), the FluPol activity was quantified in transfections containing 0 to 50 ng of the VHH16 plasmid per P96 well. An additional empty plasmid was included in each transfection to transfect the same amount of DNA. Data are mean ± s.e.m. with n = 4 technical replicates. Kruskal-Wallis test was used to compare luminescence in the presence or absence of VHHs. $p < 0.05$ is considered significant.

block virus multiplication in this assay, VHH16 displays an inhibition activity on FluPol as evidenced by a lower activity of the luciferase reporter in infected cells. We noted that VHH18 that slightly inhibited FluPol activity did not block virus replication.

To confirm this result, we aimed at determining the virus susceptibility of MDCK cell lineages constitutively expressing VHH16. Thus, a construct in which the VHH16 open reading frame was fused to sequences encoding a 2A self-cleaving peptide and GFP was placed under

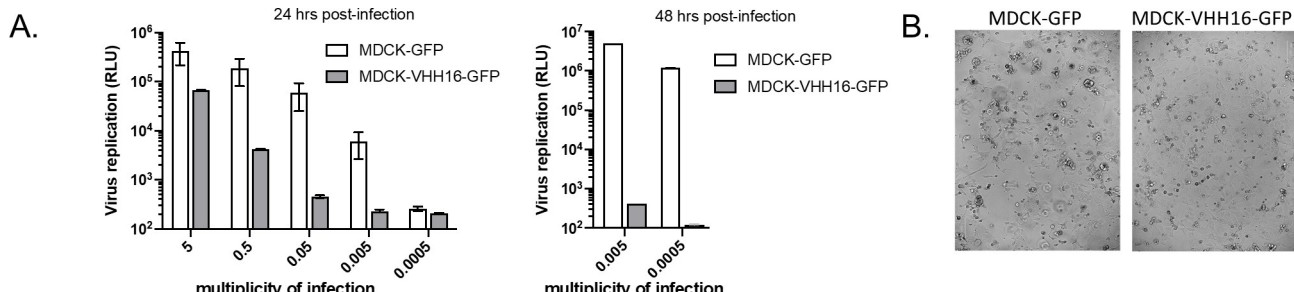

**Fig 4. MDCK cells constitutively expressing a GFP-tagged version of VHH16 (or GFP as control cells) were seeded 24 h before infection with an influenza A virus encoding a reporter nanoluciferase (WSN-Luc). (A)** Infections were carried out at different multiplicities of infection (m.o.i.) and virus replication was quantified in cell lysates 24 hours and 48 hours post-infection (n ≥ 2 replicates) **(B)** Light microscopy views of MDCK-VHH16-GFP and MDCK-GFP cells infected at a m.o.i. of 0.05 24 hours post-infection.

the control of a constitutive promoter in a lentiviral vector. MDCK cells were transduced with the resulting plasmid or with a lentivirus driving expression of GFP alone. **S4 Fig** exemplifies the stable expression of the VHH16-2A-GFP construct in selected MDCK clones. Infections of serial dilutions of the virus inoculum revealed a marked block of infection in MDCK-VHH16 cells when compared to the MDCK-GFP cells (**Fig 4A**). A $10^4$-fold decrease of virus multiplication was evidenced at 48 hrs post infection with a multiplicity of infection of 0.005. The cytopathic effect of the infection was found delayed or even blocked in MDCK-VHH16 cell monolayers, compared to MDCK-GFP cells (**Fig 4B**). Using a conventional H1N1 virus (the WSN strain) to compare virus production in MDCK-GFP and MDCK-VHH16 cells, we observed a 20-fold decrease of virus titer in the cell culture medium at 24 hrs post-infection (**S5 Fig**), confirming the inhibitory activity of VHH16.

To analyze more finely the ability of VHH16 to block influenza virus multiplication, we engineered several additional cell lines, in which the VHH16-2A-GFP construct was placed under the control of the pTRE3G promoter, which is inducible by doxycycline (**Fig 5A**). Thus, with these constructs, we may be able to quantify the neutralization activity of VHH16 when expressed at early or late stages of infection. As exemplified in **S6 Fig**, expression of VHH16-2A-GFP was revealed only under doxycycline expression induction in RK13, a rabbit influenza virus permissive cell line. To probe the effect of VHH16 expression on infection, doxycycline was added 24 hours before infection in cultured RK13-VHH16 clones. While virus multiplication was effective in uninduced cells, Dox-induced RK13 and MDCK cell clones expressing VHH16 were found refractory to influenza virus infection (**Figs 5B and S7**). A dose-response quantification of virus replication as a function of doxycycline concentration showed that 0.04 μg/ml of doxycycline blocked efficiently virus multiplication (**Fig 5C**). Next, we determined the effects of VHH16 expression when induction was carried out at different times post-infection (**Fig 5D**). Strikingly, doxycycline addition at 6 hours post-infection results in a block of 50% of virus replication.

To further investigate the spectrum activity of VHH16, we determined its neutralization activity towards two additional influenza viruses belonging to the H7N1 and H3N2 subtypes (A/Turkey/Italy/977/1999 [H7N1] and human influenza A/Scotland/20/1974 [H3N2]), and both tagged with the nanoluciferase (Nluc) reporter gene (**Fig 5E**). As found with the H1N1 virus (WSN strain), multiplication of these two last viruses was blocked efficiently when VHH16 expression was induced, suggesting sequence conservation of the VHH16 epitope. Replication of vesicular stomatitis virus (a virus unrelated to influenza viruses) was not altered after doxycycline induction.

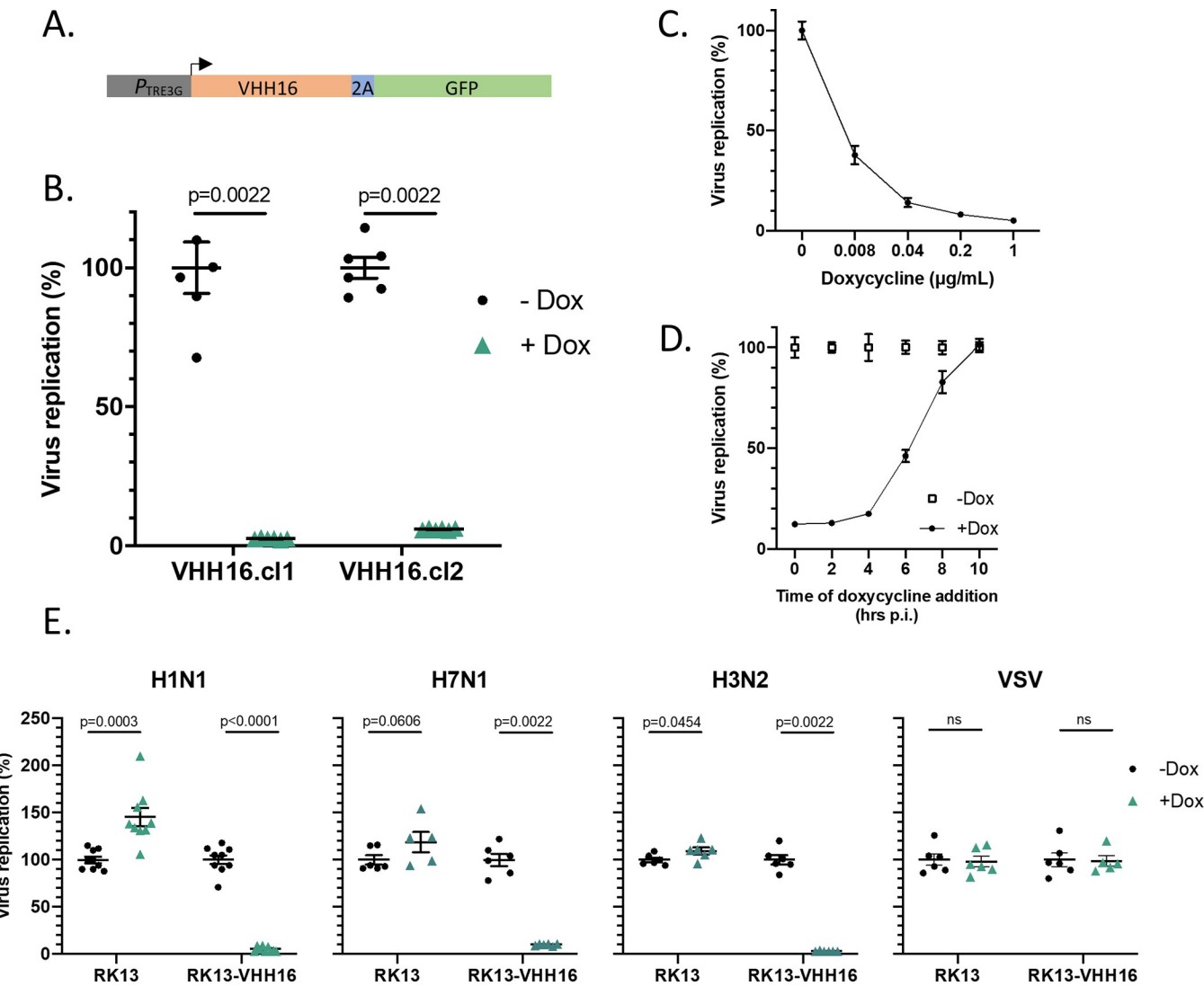

**Fig 5. Virus permissivity of RK13 cells expressing VHH16 in an inducible manner. (A)** Scheme of the DNA construct to promote VHH16-2A-GFP gene expression in a doxycycline (Dox)-inducible manner. **(B)** Two different RK13 cell clones selected for VHH16-2A-GFP gene expression were incubated (or not) with doxycycline and infected with the reporter influenza virus WSN-Luc. Twenty-four hours post-infection, virus replication was quantified by measurement of the luciferase activity. Data are mean ± s.e.m. n = 2 independent transfections with n = 3 technical replicates. 2-way ANOVA test was used to compare luminescence in the presence or absence of doxycycline. p<0.05 is considered significant. **(C)** WSN-Luc virus replication quantification as a function of Dox concentration. (n ≥ 2 biological replicates) **(D)** Quantification of WSN-Luc virus replication when Dox (1 μg/mL) was added at different times post-infection (n ≥ 2 biological replicates). **(E)** Replication quantification of influenza virus types (H1N1, H7N1 and H3N2) encoding a reporter nanoluciferase in RK13 and RK13-VHH16 cells incubated with doxycycline. VSV-dsRed virus [36] replication was quantified by measuring mCherry fluorescence in fixed cells using a TECAN spectrophotometer Infinite 200 PRO (excitation wave length 580 nm for an emission wave length lecture at 620 nm). VHH16 expression did not inhibit VSV-dsRed replication (n ≥ 2 biological replicates).

## VHH16 interacts with the PA-PB1 dimer with high affinity

To identify the FluPol subunit recognized by VHH16, we used the previously described *Gaussia princeps* luciferase-based complementation assay [23]. In this assay, an interaction between two proteins each fused to either the Luc1 or the Luc2 segments of the *Gaussia* luciferase enzyme, results in reconstitution of a functional luciferase activity, which can be quantified by addition of the substrate. To favor proper folding of the FluPol subunits and the VHH16, the two luciferase moieties were fused onto their C terminus. For each sample, the specificity of

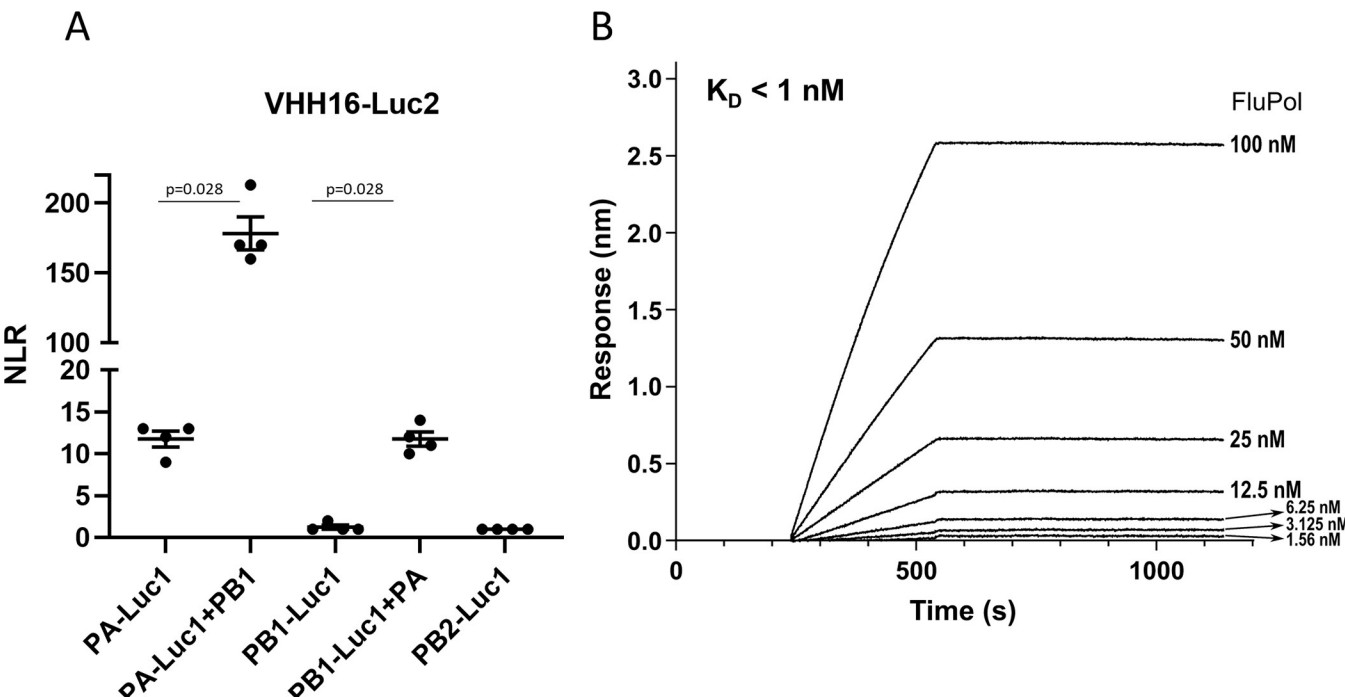

**Fig 6. The interaction between the VHH16 and FluPol.** (A) Complementation split-luciferase interaction assay between VHH16 and FluPol subunits. The normalized luminescence ratio (NLR) is calculated as described in the Materials and Methods section to quantify the interaction between VHH16 fused to Luc1 and FluPol subunits fused to Luc2. Luc1- and Luc-2 tagged polypeptides were expressed with or without untagged FluPol subunits. Twenty-four hours post-transfection, cells were lysed and luminescence was measured. Data are mean ± s.e.m. n = 4 technical replicates. Mann-Withney test was used to compare the NLR values, p<0.05 is considered significant. (B) BLI binding kinetics measurements between VHH16 and the PA-PB1 dimer. Equilibrium dissociation constants ($K_D$) was determined on the basis of fits, applying a 1:1 interaction model.

the interaction over background was estimated by calculating a normalized luminescent ratio (NLR, [24]). Co-expressed FluPol subunits PA, PB1 or PB2 (fused to Luc1) with VHH16 (fused to Luc2) revealed a NLR value of 12 for the PA-VHH16 interaction (**Fig 6A**) and no interaction between PB1 or PB2 with the VHH16. When PA-Luc1 was co-expressed with PB1 and VHH16-Luc2, NLR value reached 175, suggesting that VHH16 recognizes preferentially the PA-PB1 dimer than the monomeric PA or that PB1 stabilizes the VHH16 binding site on PA. The interaction between VHH16 and the PA-PB1 dimer was also revealed when PB1 fused to Luc1 was co-expressed with PA. VHH16 was also found to bind when the three FluPol subunits were co-expressed (**S8 Fig**). We next explore the VHH16 affinity for the PA-PB1 dimer by biolayer interferometry (BLI) at different concentrations to determine its kinetic rate constants (**Fig 6B**). VHH16 displays low on- and off-rate constants, resulting in a dissociation constant that cannot be precisely estimated, but below the nanomolar range and corresponding to an almost non-reversible interaction.

## VHH16 inhibits FluPol transcription and/or replication

As shown above, VHH16 inhibits the transcription/replication activity of FluPol when expressed without other viral components (**Fig 3**). We next confirmed in an infection assay that transcription and replication were impaired in cells constitutively expressing VHH16. Viral RNA quantification showed a decrease of mRNA, cRNA, and vRNA levels when compared to control cells (**Fig 7**). These results suggest that VHH16 neutralization activity may be due to a block of its RNA-polymerase activity in the nucleus, targeting transcription and/or

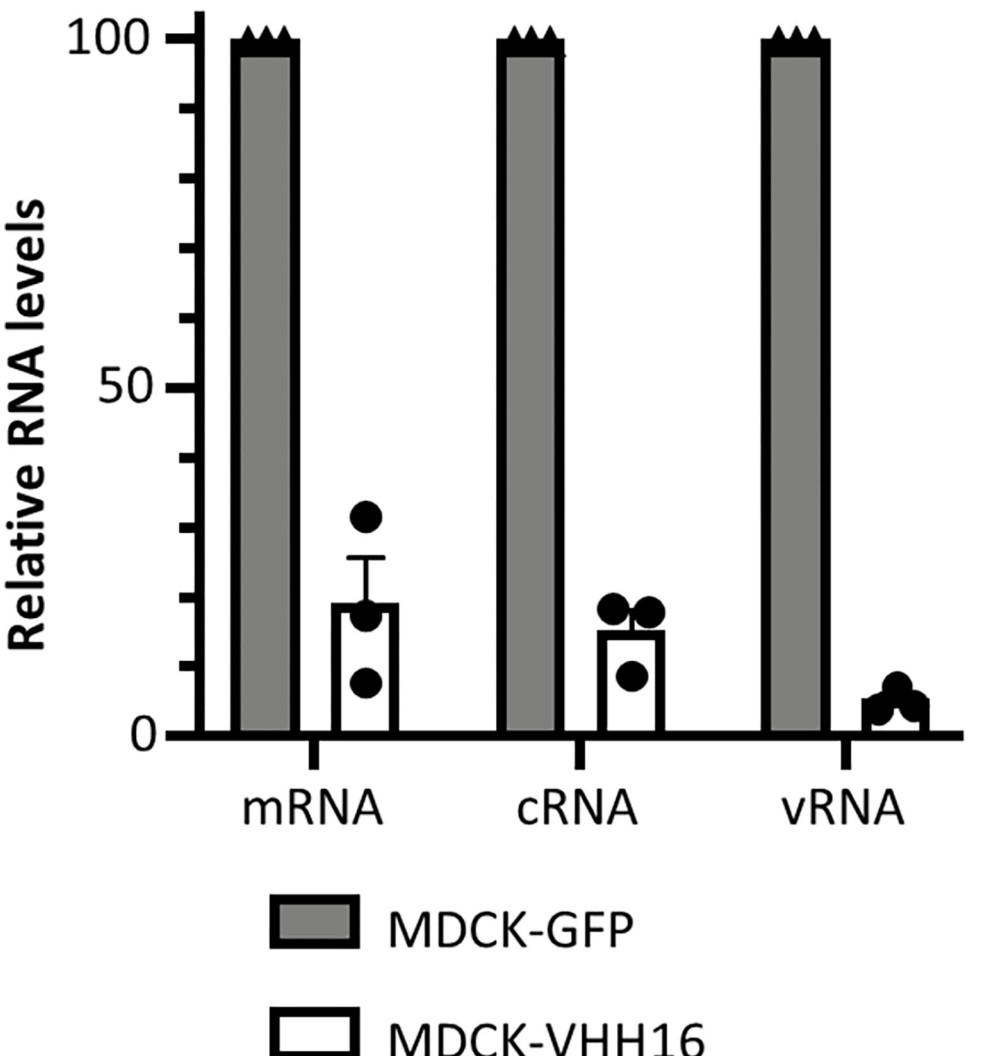

**Fig 7. Quantification of viral RNAs in WSN-infected MDCK-VHH16 cells.** Specific primers (Table 1) were used for strand-specific real-time RT-PCR using tagged primers for quantification of the vRNA, cRNA, and mRNA of segment 5.

replication, or any other earlier steps in the virus cycle such as FluPol assembly or the import of its subunits in the nucleus.

## VHH16 does not block PA-PB1 dimer / PB2 assembly

Following their synthesis in the cytoplasm, while PA and PB1 are transported in the nucleus as a dimer, PB2 is imported in this compartment separately [5], [6]. Once in the nucleus, the PA-PB1 dimer associates with PB2 to form the functional heterotrimeric polymerase. To determine whether VHH16 binding on the PA-PB1 dimer may interfere with its association of PB2 in the nucleus, we used a protein complementation assay between the PA-PB1 dimer and PB2. As shown in **Fig 8A**, while PA alone does not bind to PB2 in the absence of PB1, it associates with in the presence of PB1 (with an NLR value of about 100). Co-expression of VHH16 (or of the non-neutralizing VHH7) with the three FluPol subunits does not result in a significant decrease of the NLR, suggesting that VHH16 does not sterically hinder the PA/PB1 interaction to PB2.

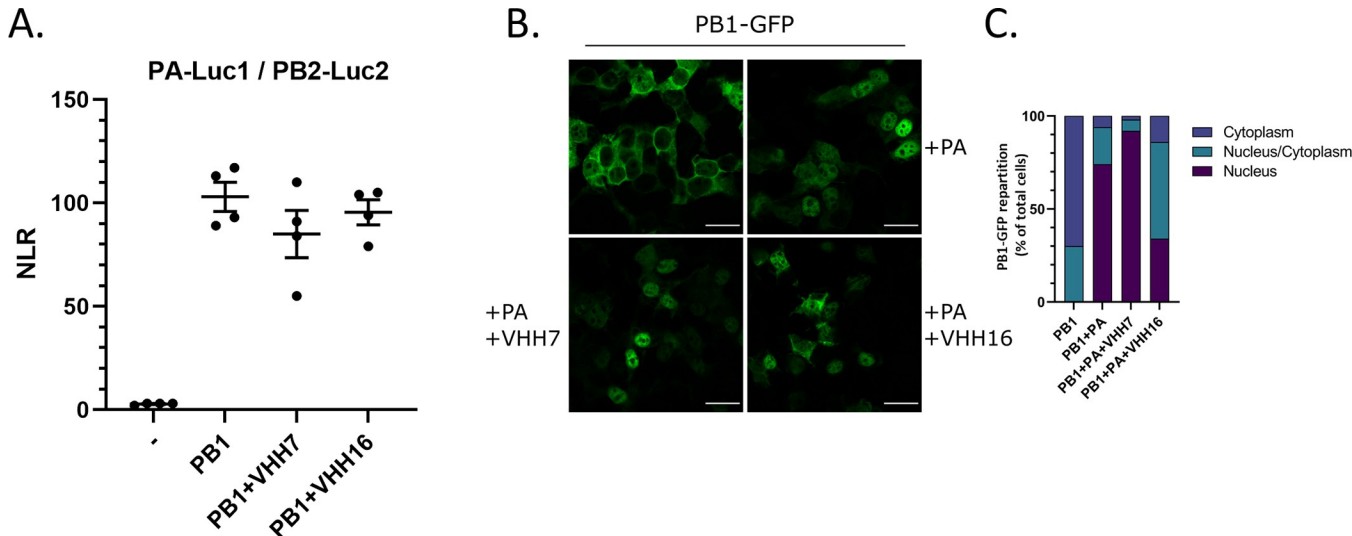

**Fig 8. FluPol trafficking and assembly in the presence of VHHs. (A)** Complementation split-luciferase interaction assay between PA-Luc1 and PB2-Luc2 fusion proteins in the presence of PB1 and VHHs as indicated. The normalized luminescence ratio (NLR) is calculated as described in the Materials and Methods section. Twenty-four hours post-transfection, cells were lysed and luminescence was measured. Data are mean ± s.e.m. n = 4 technical replicates and are representative of several experiments. **(B)** Subcellular localization of the PB1-GFP fusion protein co-expressed with PA and VHHs. **(C)** Percentage of PB1-GFP-expressing cells with nuclear versus cytoplasmic localization of PB1-GFP. Fifty GFP-positive cells were scored for each condition.

## VHH16 interferes with the transport of the PA-PB1 dimer to the nucleus

The inhibition of the FluPol RNA-polymerase activity by VHH16 could result from a block in the genome transcription/replication process or in any upstream step, such as the assembly of the polymerase subunits or the import of the polymerase subunits in the nucleus. In order to determine whether VHH16 may interfere with the nuclear import of the PA-PB1 dimer into the nucleus, we used plasmids to co-express a fluorescently labeled PB1 (PB1-GFP, [23]) and PA with VHH16 to visualize the localization of the PB1 subunit. Our experiments confirmed that when PA and PB1 were expressed together, PB1 localized into the nucleus, in contrast to what observed when PB1 was expressed alone (**Fig 8B and 8C**). Seventy-three percent of the labeled cells display an exclusive nucleus labeling when PA and PB1 were co-expressed. When PB1-GFP and PA were co-expressed with VHH16, only thirty-two percent of the cells exhibited an exclusive fluorescence signal in the nucleus. When PB1 and PA were co-expressed with VHH7 (a non-neutralizing VHH), 90% of the cells display a strict nucleus labeling. We concluded from these data that VHH16 binding interfere with the import of the PA-PB1 dimer in the nucleus. We next hypothesized that VHH16 may interfere with the binding of RanBP5, the β-importin that supports the transport of the PA-PB1 dimer to the nucleus [25].

## VHH16 does not displace the interaction of the importin RanBP5 with the PA-PB1 dimer

To determine if VHH16 may displace RanBP5 complexed with the PA-PB1 dimer, a recombinant equimolar complex made by PA and PB1 subunits with RanBP5 was purified as described previously [20] and incubated with the VHH16 to analyze its impact on the complex. Gel filtration analyses shows that VHH16 was able to bind efficiently to the PA-PB1-RanBP5 complex and did not displace RanBP5 from the PA-PB1 dimer (**S9 Fig**). This result suggests that the VHH16 binding site does not overlap the one of RanBP5 on PA-PB1.

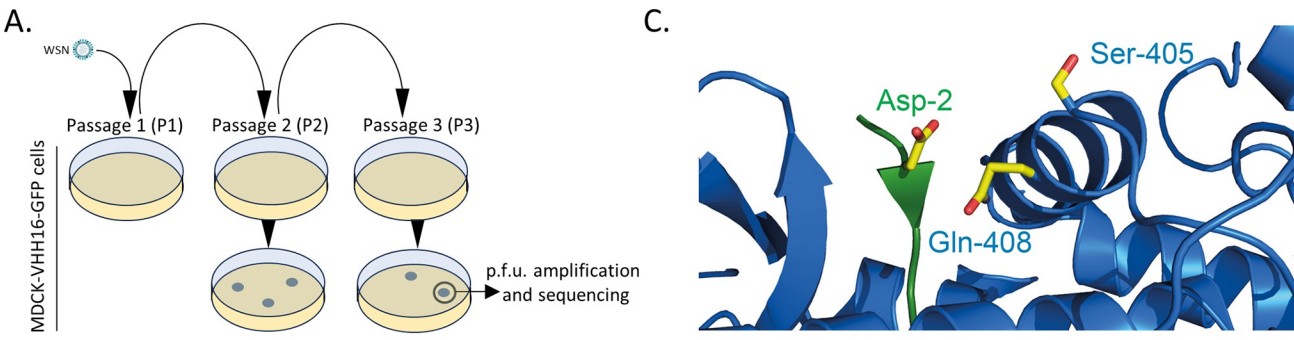

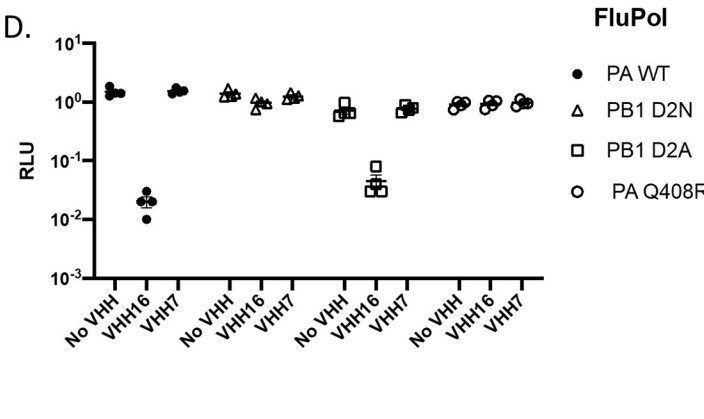

| VHH16-escape mutant | Nucleotide substitution | | Amino acid change |
|:---:|:---:|:---:|:---:|
| 1 | PA | CAG>CGG | Q408R |
| 2 | PA | AGU>AAU | S405N |
|   | PB1 | GCA>UCA | A63S |
| 3 | PB1 | GAU>GAA | D2E |
|   | PB1 | AUG>AUU | M179I |
|   | PB1 | ACA>AAC | T201N |
| 4 | PA | CAG>CGG | Q408R |
|   | PB1 | ACA>AAC | T201N |
| 5 | PA | AGU>AAU | S405N |
| 6 | PB1 | GAU>AAU | D2N |

**Fig 9. VHH16-escape mutants selection and characterization. (A)** VHH16-escape mutants screening protocol carried out in MDCK-VHH16 cells. Plaque-forming units selected after 2 or 3 passages in MDCK-VHH16-GFP cells were amplified for PA and PB1 genomic segments sequencing. **(B)** Mutations selected on PA and PB1 subunits in VHH16-escape mutants **(C)** Lateral view of the PA structure organization (in blue) around the N-terminus of PB1 (in green). The lateral chains of the amino acids that were substituted in the mutants were shown. **(D)** Effect of VHH16 expression on FluPol mutants replication/transcription activities in a luciferase-reporter minireplicon assay. Plasmids expressing the subunits of the FluPol replicative complex mutants harboring the following mutations (PA-Q408R, PB1-D2N or PB1-D2A) were co-transfected in HEK 293T cells together with the WSN-NA-firefly-luciferase reporter plasmid and with a plasmid encoding VHH16, VHH7 or with an empty plasmid (indicated as No VHH). A plasmid encoding the nano-luciferase was co-transfected to control DNA uptake and normalize minireplicon activity. Luciferase activities were measured in cell lysates 48 hours post-transfection.

### Generation and sequence characterization of VHH16-escape mutants

In order to map more precisely the VHH16 binding site on FluPol, we selected a series of VHH16-resistant variants able to multiply under the pressure of VHH16 stably expressed in MDCK cells (**Fig 9A**). Mutants were plaque-isolated after two or three serial passages in these cells. Sequencing of FluPol encoding segments of escape mutants revealed that three single amino acid changes in both segments may confer resistance to VHH16 neutralization, two in PA (S405N and Q408R) and one in PB1 (D2N) (**Fig 9B**). Interestingly, these three positions are close on the surface of FluPol and are distant of the RanBP5 binding site on PB1 defined by a large β-hairpin containing Lys and Arg residues (**Fig 9C**) [20][25]. Three additional mutants display several mutations, only in PB1 (with three substitutions in mutant 3) or in PA and PB1 (with a single mutation in each subunit in mutants 2 and 4). For each of these mutants, a

mutation in position 405 or 408 of PA, or in position 2 of PB1 (with the substitution D2E), was identified, further suggesting the importance of these residues in VHH16 binding.

## Functional validation of the VHH16 binding site

To functionally confirm the role of the substitutions PA Q408R and PB1 D2N in the VHH16-escape phenotype, we used a minireplicon assay. Thus, the RNA-polymerase activity of FluPol with mutated PA or PB1 subunits was quantified in the presence or absence of VHH16. **Fig 9D** shows that the FluPol mutants exhibit a slightly lower activity than its wild-type (wt) form, More importantly, while the activity of the wild-type form of FluPol was strongly inhibited by VHH16, the FluPol activity of its PA Q408R and PB1 D2N mutants was not inhibited, in contrast to what was observed with a non-neutralizing VHH. The FluPol PB1 D2A mutant was inhibited by VHH16, suggesting a steric hindrance in VHH16 binding to the D2N mutant. To summarize, VHH16 binding is dependent of adjacent residues located at the N-terminus of PB1 and the PA α-helix [405–415].

## Discussion

In this study, we exploited the unique properties of single-domain antibodies to identify new vulnerabilities in the functioning of FluPol. While their biosynthesis includes secretory pathway post-translational modifications, such as glycosylation and disulfide bond formation, their ability to recognize an epitope is generally well conserved when they are expressed in the cytosol of eukaryotic cells or in bacteria. We thus generated VHHs targeting FluPol and studied their impact on its activity when they were expressed in the cytosol of infected cells. While two of them were able to inhibit transcriptional/replicase activity, only VHH16 was able to block virus multiplication in a transient expression assay. Its neutralization activity was found highly potent in cell clones constitutively expressing the VHH. Moreover, VHH16 expression induction at late time of infection still resulted in a block of infection, demonstrating its potentialities. Its affinity (that was impossible to quantify precisely in the picomolar range due to a too low non-measurable off-rate constant) may account for its high neutralization activity.

### VHH16 binding site mapping

To map the VHH16 inhibitory site on FluPol, we carry out a functional screen to identify key residues in the FluPol-VHH interaction. We thus generated a series of virus mutants able to multiply in cell clones constitutively expressing the VHH. Mutants were sequenced and a functional validation of the binding site was carried out using a minigenome replication/transcription assay with FluPol mutated forms. The site bound by VHH16 is defined by PA residues on α-helix [405–415] and the adjacent PB1 N-terminal end (**Fig 9C**). Thus, the fact that VHH16-escape mutations were selected on both subunits is in full accordance with our observation that the VHH bind more efficiently the PA-PB1 dimer than to PA alone.

### Are the sequences of the VHH16 binding site conserved?

Since we identified VHH16-escape mutations relatively easily, we wanted to determine if the substitutions we revealed mark a high degree of sequence variability at these positions among influenza A viruses. While PB1 Asp in position 2 and PA Gln in position 408 are strictly conserved in the 25 sequences we choose as representatives of the influenza A diversity, position 405 appears to be slightly variable, with the identification of the conservative S405C substitution in two viruses (**Fig 10**). It can be thus assumed that the fitness of the VHH16-escape mutants may be lower than the one of the wild-type forms.

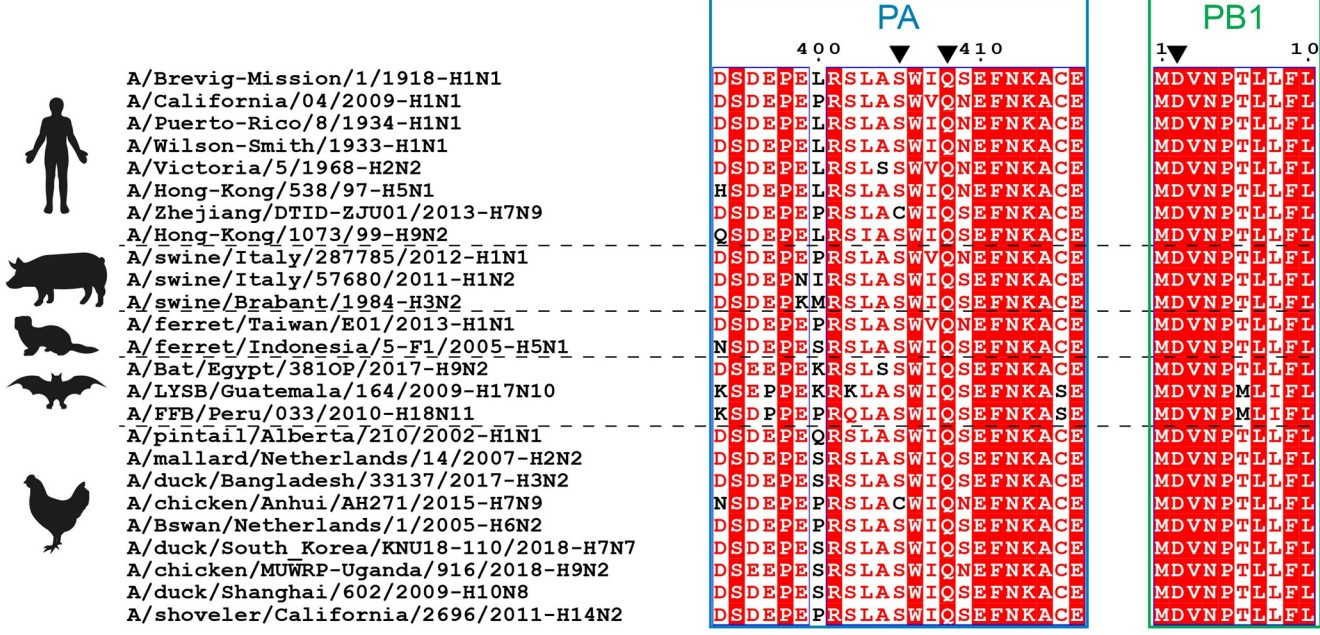

**Fig 10. Amino acid sequences alignment of 25 PA and PB1 subunits representative of the influenza A virus diversity in different hosts.** Positions in which substitutions were identified are indicated by arrowheads. Strictly conserved residues are white on a red background, and partially conserved residues are red.

Such overlap of VHH epitopes on a PA-PB1 junction has been previously recognized with other anti-FluPol inhibitory nanobodies [26]. Analyzing the sequences of these VHHs and their FluPol target sites which have been defined by cryo-EM, we identified Nb8191 as binding to the same epitope than VHH16. Interestingly, Nb8191 CDR2 loop that binds the PA α-helix [405–415] and PB1 N-terminus (PDB number: 7NIR) exhibit the same amino acid sequence than the one of VHH16 (with the CDR2 IDDGGTI amino acid stretch) (**Fig 11A**). The CDR1 and CDR3 sequences of VHH16 and Nb8191 are (highly) divergent and the ones of Nb8191 are not involved in the binding. These observations suggest that the affinities of the two VHHs for their common target may be close and that they should exhibit similar properties. Nb8191 has been recognized to significantly inhibit viral mRNA synthesis and cause reduction in virus production when transiently transfected in permissive cells [26].

## How does VHH16 inhibit virus multiplication?

Combining our functional data and the VHH16 epitope mapping, we believe that inhibition of FluPol activity may result from different specific activities of VHH16. First, our results suggest that VHH16 interferes with the import of the PA-PB1 dimer into the nucleus, this activity being not associated to a steric hindrance of the binding of the importin β RanBP5 to the dimer (the VHH16 epitope being distant from the PB1 nuclear localization signal). We favor the hypothesis that VHH16 binding may result in a slow-down of the import of the PA-PB1 complex into the nucleus, thus perturbating the virus cycle. It is interesting to note that the transport of the PA-PB1 dimer into the nucleus constitutes a critical step in temperature-sensitive influenza A virus mutants generated in the PA linker [23], [27]. In these mutants, while their production and the transport of their PA+PB1 subunits is not impaired at 33°C and 37°C, virus multiplication and the PA+PB1 transport into the nucleus is blocked at 39.5°C,

**A.**

```
                        CDR1                    CDR2                 CDR3
VHH16   QVQ..SGSVFSIDYIGWYRQAPGKQRELVATISSGGTITNY..CSANGLG--EHRVYWGQ..TVS
Nb8191  QVQ..SESVASINIVGWYRQISGKERELVARISSGGTITDY..CNAEYQYGSDWFHIWGQ..TVS
        ***  * ** **: :***** .**:***** ********:*  *.*:    :   *** ***
```

**B.**

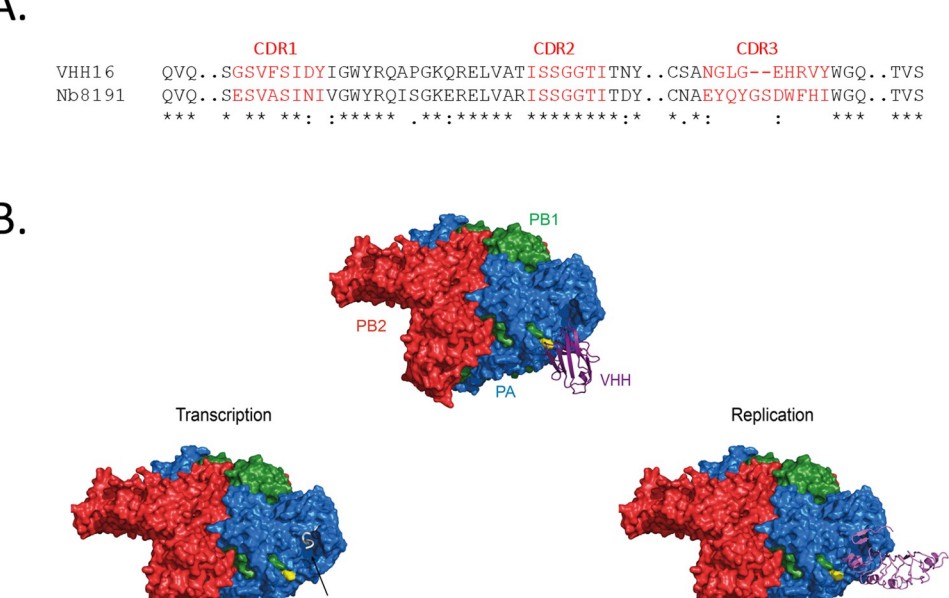

**Fig 11. 3D-model of the VHH16-FluPol interaction. (A)** Pairwise alignment of VHH16 sequence with the VHH Nb8191 previously described as specific of FluPol [26]. Note the full sequence identity in their CDR2 that constitutes the VHH paratope (PDB number 7NIR). **(B)** Surface representations of FluPol (with the N-terminus of PB1 in yellow) in complex with the VHH Nb8191 (PDB number 7NIR), RNA-Polymerase II CTD (Pol2$_{CTD}$) (PDB number 6FHH) and with ANP32A (PDB number 6XZQ). Note that the VHH binding generate a steric clash with ANP32A and RNA-polymerase II CTD.

thus confirming the vulnerability of influenza viruses at this step of the virus cycle. The position of the VHH16 binding site on FluPol strongly suggest that VHH16 interferes with viral transcription *per se*. As exemplified in **Fig 11B**, the site bound by VHH16 and Nb8191 overlaps with the binding site of the RNA-polymerase II CTD and, indeed, Nb8191 was found to inhibit the binding of FluPol to the CTD of Pol II and, consequently, inhibit transcription [26], [28], [29], [30]. VHH16 may also block viral replication. By mediating the assembly of the influenza virus replicase, host ANP32 proteins are essential for the replication of the viral RNA ([31] and references therein). Thus, an ANP32 subunit bound to two FluPol molecules forming an asymmetric dimer is believed to constitute the replication platform for the viral RNA genome. In the available cryo-EM structures of the influenza C FluPol-ANP32A complexes (PDB numbers 6XZG, 6XZPR, 6XZR and 6XZQ), the ANP32A residues [amino acids 128–130] bind the N-terminus of PB1 and the adjacent PA/P3 α-helix [405–415] (using the amino acid numbering of influenza A virus PA subunit) defining the VHH16/Nb8191 binding site (**Fig 11B**). Thus, the high neutralization activity of VHH16 may result from several complementary properties, being able to perturbate the transport of the PA-PB1 complex to the nucleus, and to block specifically viral transcription and replication (if the interaction of ANP32A is structurally conserved with influenza C and A FluPols) via steric hindrance for the binding of the CTD of RNA-polymerase II and ANP32A, respectively.

In summary, we have identified a binding site on the FluPol surface at a PA-PB1 interface that is highly sensitive to inhibition by a VHH. Since it is able to block the replication of several viruses representative of diverse influenza A subtypes, its binding site may constitute an effective target for the design of molecules with antiviral properties.

## Materials and methods

### Cells

Human embryonic kidney HEK-293T cells (ATCC) were grown in complete Dulbecco's modified Eagle's medium (DMEM, Eurobio scientific) supplemented with 10% fetal calf serum (FCS, Capricorn) and 1% penicillin-streptomycin. Madin-Darby canine kidney (MDCK) and rabbit kidney (RK13) cells were grown in modified Eagle's medium (MEM, Eurobio scientific) supplemented with 5% fetal calf serum, glutamine, and penicillin-streptomycin. For MDCK and RK13-derived clones, 4 or 1 µg/ml of puromycin was added to the cell culture medium, respectively.

### Plasmids

VHH ORFs were cloned in pCI mammalian expression vector (in Nhe1-Mlu1 restriction sites; Promega). The VHH16 ORF was also fused in frame with sequences encoding the 2A peptide of porcine teschovirus 1 and GFP and placed under the control of a constitutive promoter in a lentiviral vector–the 2A peptide allowing the co-translational cleavage between VHH16 and GFP. The VHH-2A-GFP ORF was also placed under the control of the $P_{TREG}$ promoter, which is inducible by doxycycline (Takara Bio). For minigenome assays, a pPolI-Firefly plasmid encoding the Firefly luciferase sequence in negative polarity flanked by the 5' and 3' non-coding regions of the IAV NA segment was used [23]. The pCMV-Nanoluc (Promega) plasmid was used as an internal control to normalize transfection efficiency.

### MDCK- and RK13-VHH16 cell clones selection

Cell clones were either produced by lentiviral transduction or plasmid transfection. For lentiviral production, 293T cells (ATCC) were co-transfected with pLV-EF1a-IRES-puro (Addgene 85132) encoding GFP or the fusion protein VHH16-2A-GFP, packaging plasmid R8.2 (Addgene 12263) and VSV-G plasmid (Addgene 8454) as described [32]. Supernatants containing vectors were harvested at 36h, 48h and 72h, clarified and filtered. MDCK cells in 6-wells plate were transduced for 24 hours with lentiviral vectors and puromycin selection was applied 48 hours after transduction to generate MDCK-VHH16 and MDCK-GFP cells.

MDCK and RK13 cells inducibly expressing HA-tagged VHH16 were generated by co-transfection of pCMV-TET3G, pTRE3G-VHH16 and a plasmid expressing the puromycin resistance cassette and selected in the presence of puromycin (1 µg/mL for RK13- and 4 µg/mL for MDCK- cell clones).

### Protein expression and purification

Expression and purification of the FluPol was done as previously described [20]. Briefly, large scale suspension cultures expressing the polymerase fusion constructs of A/Victoria/3/1975 (H3N2) was prepared using High Five insect cells grown in Express Five media (Life Technologies) at $0.5 \times 10^6$ cells/mL infected at 0.2% (V/V) with the baculovirus mother solution. Cultures were maintained at $0.5-1 \times 10^6$ cells/mL until proliferation arrest (24–48 h after infection). Cultures were then spun down at 800 g for 10 min and cell pellets were stored at $-80°C$.

Cell pellets were resuspended in 50 mL of lysis buffer (50 mM Tris-HCl pH 8.5, 300 mM NaCl and 2 mM β -mercaptoethanol) per $5 \times 10^8$ cells in the presence of EDTA-free anti-protease cocktail (complete from Roche). Lysis was performed with two cycles of freezing $(-180°C)$ / thawing (26°C) after which 10% of glycerol were added to the lysate before centrifugation (45 min, 40 000 g, 4°C). After retrieval of the clarified lysate, 15 mM of imidazole pH

8.0 were added before loading on a Ni-NTA Agarose column (Quiagen). After flowing the lysate through the column, wash steps were performed and bound complex was eluted with 300 mM imidazole. Elution fractions containing the polymerase core were pooled and dialysed overnight against a 20 mM Hepes pH 7.5, 300 Mm NaCl, 5 Mm betamercaptoethanol, 10% glycerol buffer before being subjected to a Hitrap heparin resin (Cytiva). After binding to the resin and wash steps, elution was carried out with a salt gradient on a FPLC system. The FluPol elution peak was pooled and flash freezing in liquid nitrogen.

## VHH library generation

Four injections of 0.5 mg purified FluPol core (in 20 mMTris–HClpH 8.0, 150 mMNaCl) were performed subcutaneously at one-week intervals followed by a fifth injection two weeks later in one llama (*Lama glama*; from Ardèche Lamas, France). Lymphocytes were isolated from blood samples obtained 5 days after the last immunization. The cDNA was synthesized from purified total RNA by reverse transcription and was used as a template for PCR amplification to amplify the sequences corresponding to the variable domains of the heavy-chain antibodies. PCR fragments were then cloned into the phagemid vector pHEN4 [33] to create a nanobody phage display library. Selection and screening of nanobodies were performed as previously described [15]. Three rounds of panning resulted in the isolation of FluPol-specific binders. VHH-7, -9, -13, -16, -18, -19 and -22 were selected and sequenced. The synthetic genes corresponding to the VHH sequences with a pelB signal peptide and a C-terminal 6xHis tag were obtained from Twist Bioscience and cloned in between *EcoRI* and *NotI* sites in a pET-28a expression vector.

## Protein complementation and minigenome assays

HEK-293T cells were seeded in 48 well plates ($1x10^5$ cells/well) one day before being transfected with 150ng pci-VHH16-Luc2 together with 150ng of either pci-PA-Luc1, pci-PB1-Luc1 or pci-PB2-Luc1 and another 150ng of either pci-neo, pci-PB1 or pci-PA using polyethylenimine (PEI). 24h later, cells were lysed using Renilla lysis buffer (Promega) for 20min at room temperature under shaking (250 rpm) and luminescence was quantified with the Renilla Luciferase Assay System (Promega).

HEK-293T cells were seeded in 96 well plates ($5x10^4$ cells/well) one day before being transfected with 25 ng of each of the pCI plasmids expressing the PB2, PB1, PA viral proteins, together with 50, 10 and 5ng of the pCI-NP, pPolI-Firefly and pCMV-Nanoluc plasmids, respectively, and 10ng or a range from 0 to 50ng pci-VHH using PEI. 48h later, cells were lysed using PLB buffer (30 mM Tris pH7.9, 10 mM $MgCl_2$, 1% Triton X-100, 20% Glycerol, 1 mM DTT) buffer and luminescence activity was measured with the Luciferase Assay System (Promega).

## Viruses

The recombinant reporter virus WSN(H1N1)-nanoluciferase (H1N1-Nluc, previously named PASTN or PA-SWAP-2A-NLuc50 in [22]) was kindly provided by Andrew Mehle. In this reporter virus, the "self-cleaving" 2A peptide from porcine teschovirus and the nanoluciferase coding sequence were placed downstream of the PA sequence to create a contiguous ORF. Native packaging sequences were restored by repeating the terminal 50 nucleotides of the PA ORF (including the stop codon) after the Nluc stop codon adjacent the native untranslated region. Direct repeats were removed from the reporter gene by introducing silent mutations at the 3' end of the PA ORF. Reverse genetic systems for avian influenza A/Turkey/Italy/977/1999 [H7N1] and human influenza A/Scotland/20/1974 [H3N2] viruses have been previously

elaborated and used to generate the recombinant reporter viruses H7N1-Nluc and H3N2-Nluc [27], [34]. Both these Nluc reporter viruses were designed using the same strategy than H1N1-Nluc, with their PA segments encoding a PA-2A-Nluc polyprotein, and were kindly provided by Ronan Le Goffic. The reporter VSV-mCherry virus [35] in which the fluorescent protein mCherry ORF was inserted in the L-protein reading frame was kindly provided by Emmanuel Heilmann.

## Infection assays

**In cells transiently expressing VHHs.** HEK-293T cells were seeded in 6-well plates at a density of $0.8 \times 10^6$ cells. Cells were transfected with 2,5µg of pCI plasmid encoding VHH7, VHH9, VHH16, VHH18 or an empty pCI plasmid, using PEI. Subsequently, 20h post-transfection, cells were infected with a WSN-PA-nanoluc virus at a multiplicity of infection of 0,01. Cells were lysed 24h after, using PLB buffer and luminescence activity was measured with the Nano-Glo Luciferase Assay System (Promega).

**In MDCK and RK13 cell clones.** Cells were seeded in 12-well plates at a density of $0.3 \times 10^6$. In (Dox+)-inducible cells, doxycycline was added at 1µg/mL (or at the indicated concentration) 24h before infection to induce VHH expression. Cells were infected with influenza viruses tagged with a reporter gene (nanoluciferase) or by VSV-mCherry at a multiplicity of infection of 0.001. Virus replication was quantified by measuring nanoluciferase luminescence and mCherry fluorescence activities.

## Immunofluorescence assay

HEK 293T cells were seeded in 24-well plates on coverslips 24 hours before being transfected with 600ng pci-PB1-GFP, pci-PA and pci-VHH16 total. After 24 hours, cells were fixed with 4% paraformaldehyde (PFA) and VHH16 was revealed using a secondary antibody coupled to AlexaFluor 546. Coverslips were then visualized by confocal microscopy.

## Affinity determination by Biolayer interferometry

Binding kinetics experiments were performed on an Octet system (Octet RED96) (FortéBio, CA). A black bottom 96-well microplate was filled with 200 µL of solution (PA-PB1 dimer in PBS buffer) and agitated at 1,000 rpm, and all experiments were carried out at 25˚C. Tips were hydrated in PBS buffer for 1 hour at room temperature prior experiments. Biotinylated VHH16 (4 µg/mL) were loaded on streptavidin SA (18–0009) biosensors (Pall ForteBio) for 1 min. After a baseline step in assay buffer (PBS [pH 7.4], 0.1% bovine serum albumin, 0.02% Tween 20), ligand-loaded sensors were dipped into known concentrations of PA-PB1 dimer for an association phase during 500 to 700 seconds. The sensors were then dipped in assay buffer for a dissociation step during 1000 seconds in assay buffer. Association and dissociation curves were globally fitted to a 1:1 binding model. Binding curves were fit using the "association then dissociation" equation in the FortéBio Data analysis software version 7.1 to estimate the $K_D$.

## Interaction assays by size exclusion chromatography

The heterotrimeric complex $PA_{200-714}$-$PB1_{1-686}$-RanBP5 was purified as previously described [20]. Size exclusion chromatography (SEC) experiments to assess the complex with VHH-16 were performed in the buffer (20 mM Hepes pH 7.4, 150 mM NaCl, 0.5% glycerol and 2 mM β-mercaptoethanol) using a Superdex 200 increase 10/300GL column (GE-Healthcare) connected to a NGC system (Bio-Rad). $PA_{200-714}$-$PB1_{1-686}$-RanBP5 (4.3 µM) was incubated with or without 9 µM VHH-16 in 700 µL for 1 hour at room temperature before injection.

**Table 1. Primer sets for strand-specific real-time RT-PCR.**

| Target | | Purpose | Primer name | Sequence (5'->3') |
|---|---|---|---|---|
| Segment 5 WSN | vRNA | Reverse transcription | vRNAtag_740F | GGCCGTCATGGTGGCGAAT GAATGGACGGAGAACAAGGATTGC |
| | | Real-time PCR | vRNAtag | GGCCGTCATGGTGGCGAAT |
| | | | 840R | CTCAATATGAGTGCAGACCGTGCT |
| | cRNA | Reverse transcription | cRNAtag_1565R | GCTAGCTTCAGCTAGGCATC AGTAGAAACAAGGGTATTTTTCTTT |
| | | Real-time PCR | cRNAtag | GCTAGCTTCAGCTAGGCATC |
| | | | 1466F | CGATCGTGCCCTCCTTTG |
| | mRNA | Reverse transcription | mRNAtag_dTR | CCAGATCGTTCGAGTCGT TTTTTTTTTTTTTTTTCTTTAATTGTC |
| | | Real-time PCR | mRNAtag | CCAGATCGTTCGAGTCGT |
| | | | 1466F | CGATCGTGCCCTCCTTTG |

## Viral RNA quantification by RT-PCR

Strand-specific real-time RT-PCR was carried out as previously described using primers specific of segment 5 [36]. The primers used are listed in **Table 1**. Briefly, cDNAs were synthesized with strand-specific RT primers tagged at their 5′ end with the hot-start modification of using saturated trehalose: a 5.5 μL mixture containing 200 ng of total RNA sample and 10 pmol of tagged RT primer was heated for 10 min at 65˚C, chilled immediately at 4˚C for 5 min, and then heated again at 60˚C. After 5 min, 14.5 μL of a preheated reaction mix (4 μl First Strand buffer 5X (Thermo Fisher), 1 μL 0.1 M dithiothreitol (DTT), 1μL dNTP mix (10 mM each), 1μL Superscript III reverse transcriptase (200 U/μl, Thermo Fisher), 1 μL RNasin Plus RNase inhibitor (40 U/μL, Promega) and 6.5 μL saturated trehalose) was added and incubated at 60˚C for 1 h. Real-time PCR (qPCR) was performed with the iTaq Universal SYBR Green Supermix (BioRad) on a CFX96 (BioRad). Seven microliters of a 10-fold dilution of the cDNA was added to the qPCR reaction mixture (10 μL Brilliant II SYBR Green qPCR Master Mix 2X, 1.5 μL of each qPCR primer at 10 μM). The cycle conditions of qPCR were 95˚C for 5 min, followed by 50 cycles of 95˚C for 15 s, 56˚C for 15 s and 60˚C for 45 s.

## VHH16-escape virus mutants selection and nucleotide substitutions identification

VHH16-escape virus mutants were selected as described in **Fig 9A**. MDCK-VHH16 cells were seeded in 12-wells plates and infected at various multiplicity of infection with the H1N1 WSN strain. Cell culture media were collected when a cytopathic effect was observed and further reported on fresh MDCK-VHH16 cells for a second and a third round of selection. To isolate virus mutants, MDCK-VHH16 cells plated in 6-wells plates were infected with passages 2 or 3 cell culture supernatants and overloaded with an agar suspension. Virus clones were picked from the agar, resuspended in cell culture medium for amplification in VHH16-expressing cells. Total RNA from cells infected with virus mutants was extracted with Trizol reagent and full segments encoding PA, PB1 and PB2 were amplified thanks to specific RT-PCR (**Table 2**).

**Table 2. Primer sets for RT-PCR of FluPol genomic segments.**

| | Sequence (5'->3') | | Sequence (5'->3') | | Sequence (5'->3') |
|---|---|---|---|---|---|
| PA.U12 | AGCGAAAGCAGGTACTGA | PB1.U12 | AGCGAAAGCAGGCAAACC | PB2.U12 | AGCGAAAAGCAGGTCAAT |
| PA.1495rev | TGGTCTTTCGCCTTCCCTC | PB1.1841rev | CCCATTTCAAGCAGACTTCAGG | PB2.1743rev | GTACAGCATTGTAGG |
| PA.1426for | GCCTTGCTTAATGCATCCTG | PB1.1409for | GAGTCAACAGGTTTTATCG | PB2.1422for | CGACATGACTCCAAGCAC |
| PA.2130rev | TCTCAATGCATGTGTGAGG | PB1.2300rev | CATGAAGGACAAGCTAAATTC | PB2.2307rev | CGTTTTTAAACTATTCGACAC |

PCR products were submitted to Sanger DNA sequencing. Sequences were analyzed with Serial Cloner and compared with the original WSN sequences. Six mutants were identified and numbered from 1 to 6. Identified nucleotide substitution were introduced in pcDNA3-PA or -PB1 thanks to site-directed mutagenesis for functional characterization in a minigenome assay.

## Supporting information

**S1 Fig. Effect of intracellular VHH expression on FluPol activity.** Plasmids expressing NP, PA, PB1, PB2 of the WSN (H1N1) strain were co-transfected in HEK-293T cells together with the NA-firefly-luciferase reporter plasmid and a plasmid encoding a VHH or an empty plasmid (indicated as No VHH). A plasmid encoding the nano-luciferase was co-transfected to control DNA uptake and normalize minireplicon activity. Luciferase activities were measured in cell lysates 48 hours post-transfection. Colors indicate distinct biological replicates.
(TIF)

**S2 Fig. VHHs expression in HEK-transfected cells revealed by indirect immunofluorescence using an anti-HA-tag antibody.**
(TIF)

**S3 Fig. Effect of the VHHs transiently expressed on the replication of WSN(H1N1)-Nluc influenza virus in HEK 293T cells.** Data are mean ± s.e.m. n = 2 independent transfections with n = 3 technical replicates. Matt-Whitney test was used to compare replication in the presence and absence of VHHs at 24 hours post-infection.
(TIF)

**S4 Fig. Visualization of the green fluorescent protein (GFP) in transduced MDCK clones stably expressing GFP (MDCK-GFP) and VHH16-2A-GFP (MDCK-VHH16).**
(TIF)

**S5 Fig. Infectious virus production in MDCK clones stably expressing VHH16-2A-GFP or GFP.** Cells were infected with the H1N1 WSN strain at a multiplicity of infection of 1. Virus production was measured 24 hours post-infection by plaque formation in MDCK cells.
(TIF)

**S6 Fig. Visualization of the green fluorescent protein in RK13 clones expressing VHH16-2A-GFP in a doxycycline-inducible manner.**
(TIF)

**S7 Fig. Subcellular localization of VHH16-HA in MDCK clones expressing VHH16-HA-2A-GFP in a doxycycline-inducible manner.** B. Two different MDCK cell clones selected for VHH16-2A-GFP gene expression were incubated (or not) with doxycycline and infected with the reporter influenza virus WSN-Luc. Twenty-four hours post-infection, virus replication was quantified by measurement of the luciferase activity. Data are mean ± s.e.m. n = 2 independent transfections with n = 3 technical replicates.
(TIF)

**S8 Fig. Complementation split luciferase interaction assay between VHH16 and the PA subunit.** The normalized luminescence ratio (NLR) is calculated as described in the Materials and Methods section to quantify the interaction between VHH16 fused to Luc1 and PA fused to Luc2. VHH16 and PA were expressed with or without PB1 and PB2 subunits. Twenty-four hours post-transfection, cells were lysed and luminescence was measured. Data are mean ± s.e.

m. n = 4 technical replicates.
(TIF)

**S9 Fig. Evidence for a complex constituted by PA$_{200-714}$-PB1$_{1-686}$-RanBP5 with VHH16.**
Purified PA$_{200-714}$-PB1$_{1-686}$-RanBP5 assemblies were incubated with VHH16. Gel filtration
chromatograms of PA$_{200-714}$-PB1$_{1-686}$-RanBP5 (A, left panel) and PA$_{200-714}$-PB1$_{1-686}$-RanBP5
incubated with VHH16 (B, left panel) were shown. Aliquotes of fractions were analyzed by
SDS-PAGE and Coomassie blue staining (right panels). VHH16 was found to stably bind the
PA$_{200-714}$-PB1$_{1-686}$-RanBP5 assemblies.
(TIF)

## Acknowledgments

We thank Andrew Mehle for the gift of the reverse genetics system of the IAV H1N1-Nluc,
Ronan Le Goffic for the gift of the H7N1-Nluc and the H3N2-Nluc viruses. We thank Nicolas
Meunier for his help in statistical analyses. This work used the platforms of the Grenoble
Instruct-ERIC Center (ISBG: UMS 3518 CNRS-CEA-UGA-EMBL) with support from FRISBI
(ANR-10-INBS-05-02) and GRAL (ANR-10-LABX-49-01) within the Grenoble Partnership
for Structural Biology (PSB). Authors acknowledge the SPR/BLI platform scientific responsi-
ble, Jean-Baptiste Reiser Ph.D., for its help and assistance.

## Author Contributions

**Conceptualization:** Bernard Delmas.

**Data curation:** Mélissa Bessonne, Magali Grange, Bernard Delmas.

**Formal analysis:** Mélissa Bessonne, Jessica Morel, Quentin Nevers, Magali Grange, Alain
Roussel, Thibaut Crépin, Bernard Delmas.

**Funding acquisition:** Bernard Delmas.

**Investigation:** Mélissa Bessonne, Jessica Morel, Quentin Nevers, Bruno Da Costa, Allison Bal-
landras-Colas, Magali Grange, Alain Roussel, Thibaut Crépin.

**Methodology:** Mélissa Bessonne, Jessica Morel, Quentin Nevers, Allison Ballandras-Colas,
Magali Grange, Alain Roussel, Thibaut Crépin.

**Resources:** Alain Roussel, Bernard Delmas.

**Supervision:** Alain Roussel.

**Visualization:** Mélissa Bessonne, Florian Chenavier.

**Writing – original draft:** Bernard Delmas.

**Writing – review & editing:** Mélissa Bessonne, Quentin Nevers, Alain Roussel, Thibaut Cré-
pin, Bernard Delmas.

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
