## [Decision Letter · Decision Letter 0]

31 Oct 2023

Dear Dr Delmas,

Thank you very much for submitting your manuscript "Antiviral activity of intracellular nanobodies targeting the influenza virus RNA-polymerase core" for consideration at PLOS Pathogens. As with all papers reviewed by the journal, your manuscript was reviewed by members of the editorial board and by several independent reviewers. In light of the reviews (below this email), we would like to invite the resubmission of a significantly-revised version that takes into account the reviewers' comments.

The reviewers found the work  interesting, but identified a number of weaknesses including overinterpretation of the data, and the lack of structural insights into the VHH16 interaction with the components of the viral polymerase.

We cannot make any decision about publication until we have seen the revised manuscript and your response to the reviewers' comments. Your revised manuscript is also likely to be sent to reviewers for further evaluation.

Sincerely,

George A. Belov, PhD

Academic Editor

PLOS Pathogens

Meike Dittmann

Section Editor

PLOS Pathogens

Kasturi Haldar

Editor-in-Chief

PLOS Pathogens

orcid.org/0000-0001-5065-158X

Michael Malim

Editor-in-Chief

PLOS Pathogens

orcid.org/0000-0002-7699-2064

Reviewer's Responses to Questions

**Part I - Summary**

Reviewer #1: In this manuscript, Bessonne and colleagues have generated seven nanobodies against the influenza A virus polymerase using a complex of the PB1 polymerase subunit and the C-terminal domain of the PA polymerase subunit as target. They found that two of these nanobodies inhibited polymerase activity in a minigenome assay, and one of them inhibited influenza virus replication in cell culture. The authors attribute this inhibitory effect to a block in the nuclear import of the PB1-PA dimer.

While the manuscript presents some intriguing data, the authors appear to miss the opportunity to fully capitalize on their initial success in developing nanobodies targeting the influenza virus polymerase. While they convincingly demonstrate that VHH16 effectively inhibits viral replication and the data are consistent with VHH16 impeding the nuclear import of PB1-PA, the conclusion that VHH16 does not inhibit RanBP5 or PB2 binding is less strongly supported (see specific points). The study’s impact could be substantially enhanced by presenting structural models to reveal where the inhibitory and non-inhibitory nanobodies bind PB1-PA.

Reviewer #2: In this work the authors identified nanobodies targeting the influenza polymerase from PBMCs of immunized llamas. The authors then tested the most potent nanobody, VHH16, in cell lines for anti-influenza activity. Overall the manuscript is neatly written and experiments were appropriately designed. However, given the level of completeness of the work, the reviewer struggled to accept the manuscript for publication in PLOS Pathogens. There are also some concerns on data quality, as I shall elaborate below.

Reviewer #3: The manuscript by Bessonne et al., investigates the mechanism of inhibition of viral replication by VHH16, a llama nanobody raised against the core of the influenza A/H3N2 polymerase (PA/197-716+PB1+PB2/1-116). Using both the minigenome assay and virus infected cells, they show that intracellularly expressed VHH16 potently inhibits viral RNA synthesis and viral replication (even when induced 6 hours post infection) and this holds for various FluA subtypes (H1N1, H7N1, H3N2) but not FluB. They further show by split luciferase complementation that interacts with PA alone but not PB1 or PB2. However, the interaction with the PA-PB1 heterodimer is much stronger (in the nanomolar range). Using the same assay, VHH16 does not inhibit PB2 assembling with PA-PB1. However, the nanobody leads to reduced (but not eliminated) nuclear import of the PA-PB1 heterodimer whilst not preventing the interaction with RanBP5, its nuclear import receptor.

The authors do not identify the epitope of VHH16, nor the exact mechanism of replication inhibition (since heterotrimeric polymerase does still form in the nucleus), but suggest that its epitope could be an affective target for antiviral therapy.

The data in the paper is generally convincing and supports the conclusions. However, while it is interesting to have generated such a nanobody, the manuscript would be much stronger if it included the proposed structural studies to identify the epiotope, which by cryoEM should be not too difficult.

**Part II – Major Issues: Key Experiments Required for Acceptance**

Reviewer #1: 1. Fig. 3A. The authors should include Western blot data to demonstrate that the nanobodies are expressed in HEK-293T cells. The lack of inhibition observed for the remaining five nanobodies could be due to lack/reduced expression. Nanobody expression should also be demonstrated for the infection experiment in Fig. S2.

2. Fig. 3D. The rationale of this experiment is poorly explained. If the authors’ aim was to address whether VHH16 affects transcription or replication or both a more appropriate assay would have been to analyse the effect on primary and secondary transcription and genome replication during viral infection. Instead, the authors use a polymerase mutant supposedly deficient in replication, but it is not shown whether this mutant is replication-deficient in the experimental setting used here.

3. Replicates and statistics. Numbers of biological and technical replicates need to be specified (Fig. 4A, Fig. 5C, D, E, Fig. S6) and error bars should be defined (Fig. 4A); please check the calculation of p values in Fig. 3C, D (all set to p=0.028), Fig. 5B, E (p=0.0022) and Fig. 6A (p=0.028). In Fig. 3A technical replicate data points from the different transfections should be distinguished. In Fig. 6A and 7A specify number of experiments; “representative of several experiments” (lines 513 and 524) is too vague.

4. Fig. 6A and 7A. These experiments require Western blots to show the expression levels of polymerase subunits (both tagged and untagged). Otherwise, the authors cannot conclude that VHH16 preferentially recognises the PB1-PA dimer. The increased value might simply reflect the increased levels of PA expression in the presence of PB1. The discussion around the more efficient binding of VHH16 to the PB1-PA dimer (lines 251-261) is only relevant if the authors show data that the observed enhanced interaction is not due to higher amounts of PA-Luc1 being present. Also, it would have been potentially informative to include data for interaction between VHH16-Luc2 and PA-Luc1+PB1+PB2 addressing whether PB2 influences nanobody binding.

5. Fig. 7D. While this experiment is suggestive that VVH16 binds the PB1-PA-RanBP5 complex, in the absence of further controls it is not conclusive. Presumably VVH16 is absent from the 9.8-12.4 ml fractions if no PB1-PA-RanBP5 is present but this is not shown. Correct FluPol-RanBP5 to PB1-PA-RanBP5 (line 531).

Reviewer #2: (1) In Figure 3, a minireplicon assay was employed to quantify effects of different VHHs. The reviewer finds it uncomfortable to accept the results with the large error in each set. Besides, the authors should quantify the expression levels of each RNP component and the co-expressed VHH in each set by Western blotting. A protein co-expressed at high expression level may (in my experience) sometimes exhaust the cells. Hence, it is necessary to include a proper control (an empty vector or, best, a plasmid expressing an irrelevant VHH), which is not seen here.

(2) Figure 5: The authors used a recombinant virus expressing a PA-NLuc fusion as reporter. Theoretically this should work well in most circumstances. However in this particular case it would be difficult to validate the applicability of this virus when the nanobody actually targeted the same PA protein. The authors are suggested to validate the viral titre with TCID50 assay or standard plaque assay.

(3) The authors did not demonstrate (or at least discuss) clearly the mechanism involved. The facts that PA-PB1 nuclear import being impaired and PA-PB1-RANBP5 not being interupted seem contradictory. This follows that the discussion on the mechanism should be rewritten to give more evidence-supported claims instead of putting forward too many guessings (e.g. the sentences about NP look irrelevant).

(4) Related to (3), Figure 7D is not clearly presented and not stated in the Methods section. If this is from a gel filtration experiment, the elution profile (chromatogram) should be provided. Besides, when a discrepancy as stated in (3) occurs, it is worthwhile to validate this result in cells (e.g. by co-immunoprecipitation) using full-length PA instead of purified proteins.

PS -- there is a large input band for VHH16 (lane 1). However the eluted VHH bands look weaker than expected. Was the binding of VHH16 not as efficient in this system as required to displace/block RANBP5?

(5) There has been reports of nanobodies targeting surface proteins of flu virus (HA/NA) which produced good potencies. These nanobodies can reach the target binding site on cell surface. However few reports nanobodies targeting internal proteins of the virus. All experiments in the manuscript involved the use of transfected plasmids to express the nanobody which is not practically feasible in animal experiments. Hence the authors are recommended to perform in vivo experiments (maybe by infecting mice with AAVs?) to demonstrate that these nanobodies can reach their target proteins in vivo and elicit their effects.

Reviewer #3: Structural studies should be pursued to identify the epitope and yield more light on the mechanism of inhibition.

**Part III – Minor Issues: Editorial and Data Presentation Modifications**

Reviewer #1: 6. Fig. 2B. There are no residues white on a red background in the figure (lines 160-161).

7. Fig. 2S. Why only a selection of nanobodies was tested in the virus inhibition assay? Nanobodies with no detectable effect on polymerase activity could still inhibit virus replication be interfering with a step downstream of viral RNA synthesis and could be of interest. Also, please speculate why VHH18 that was found to inhibit polymerase activity had no impact on viral replication.

8. Fig. 5C and line 157. Clarify units of doxycycline concentration (uM in the text vs ug/mL in the figure). Include information on the time of doxycycline treatment and infection.

9. Fig. 7B, C. The description of experimental conditions needs improvement. Which cells were used in this experiment and at what time point was GFP analysed? How many replicates have been performed? The data in panel C seem to represent data from a single experiment. Y axis is not defined.

10. Lines 244-246. The data show that only two nanobodies inhibited reporter gene expression significantly.

11. Lines 247-249. Please speculate on how the block of infection at late time points fits with the proposed mode of action of limiting the PB1-PA import into the nucleus. Also, please speculate how VHH16 could inhibit nuclear import without affecting RanBP5 binding.

Reviewer #2: Methods: Bio layer  Biolayer?

Reviewer #3: (1) P6, line 104. Here it says that the llama was immunised with WSN core but in the methods and reference 20 it says the construct was the H3N2 core.

(2) P9, line 182, Figure 6A. Can the authors explain why the NLR with PB1-Luc1+PA was only 10, whereas that with PA-Luc1+PB1 was 175? It should be stated whether Luc1 was on the N- or C-ter of PA and PB1?

(3) P10, line 212. The split luciferase complementation suggests that PA, PB1 and PB2 can assemble as a heterotrimer in the presence of VHH16. This is not surprising as the antigen used to generate the nanobody included the major stable PB1-PB2 interface (PB1-Cter with PB2-Nter). The experiment does not determine whether the presumed VHH16 bound heterotrimer is functional in binding the vRNA promoter or in RNA synthesis. Does the nanobody bind to the full recombinant trimeric polymerase in vitro?

PLOS authors have the option to publish the peer review history of their article (what does this mean?). If published, this will include your full peer review and any attached files.

Reviewer #1: No

Reviewer #2: No

Reviewer #3: No

Figure Files:

Data Requirements:

Please note that, as a condition of publication, PLOS' data policy requires that you make available all dat

---

## [Decision Letter · Decision Letter 1]

16 Apr 2024

Dear Dr Delmas,

Thank you very much for submitting your manuscript "Antiviral activity of intracellular nanobodies targeting the influenza virus RNA-polymerase core" for consideration at PLOS Pathogens. As with all papers reviewed by the journal, your manuscript was reviewed by members of the editorial board and by several independent reviewers. The reviewers appreciated the attention to an important topic. Based on the reviews, we are likely to accept this manuscript for publication, providing that you modify the manuscript according to the review recommendations.

Please address the remaining reviewers' concerns, and in particular include in the discussion the results from a recnt characterization of a similar nanobody (PMID: 35017564)

Sincerely,

George A. Belov, PhD

Academic Editor

PLOS Pathogens

Meike Dittmann

Section Editor

PLOS Pathogens

Michael Malim

Editor-in-Chief

PLOS Pathogens

orcid.org/0000-0002-7699-2064

Reviewer Comments (if any, and for reference):

Reviewer's Responses to Questions

**Part I - Summary**

Reviewer #1: This is a revised version of a manuscript by Bessonne and colleagues considered earlier for publication in PLoS Pathogens. The authors have substantially improved their manuscript. Most importantly, using passaging experiments of influenza A virus in the presence of the inhibitory nanobody and identifying mutants that escape inhibition, they now provide convincing evidence that the nanobody binds at the interface of PA and the N-terminus of PB1.

However, some issues still remain that should be addressed.

Reviewer #2: This is a revised manuscript submitted by Bessonne and co-workers. The authors have enriched the manuscript by providing more mechanistic details of VHH16 action, a nanobody identified by the team to be active in suppressing influenza polymerase activity. As such, the data now better support the conclusion. The data quality has also improved. I recommend acceptance of the manuscript upon the minor revision stated in part III.

Reviewer #3: The revised manuscript has satifactorally addressed/corrected a number of issues raised by the various referees. Furthermore, the revised manuscript is strengthened by identification of the binding site of VHH16 via generation of escape mutants, which enables a plausible explanation of the inhibitory effect to be deduced as it overlaps the Pol II CTD and (presumed, based on the FluC structure) ANP32 binding sites.

On the other hand, the fact that a similar nanobody, with an identical CDR2 binding loop, has already been functionally and structurally characterised (Nb8191, Keown et al, 2022) now considerably detracts from the novelty of this study.

**Part II – Major Issues: Key Experiments Required for Acceptance**

Reviewer #1: None

Reviewer #2: Nil.

Reviewer #3: (No Response)

**Part III – Minor Issues: Editorial and Data Presentation Modifications**

Reviewer #1: Fig. 7B: The authors have failed to address the criticism of their use of the PA-C95A mutant as a replication-deficient mutant without providing any data that this mutant is actually replication deficient in the experimental setting they use. Since this experiment provides little information on whether transcription or replication is affected, I would suggest deleting these data. In the light of the new data it is highly likely that the nanobody inhibits both transcription and replication.

Fig. 9C: Please provide the PDB ID for the structure.

Fig. S5: Please provide information on the virus used, the MOI, and the timepoint in the legends. What additional information does this figure provide compared to the data in Fig. 4?

Fig. S9: The panels and legends lack information; define the blue and red curves, label bands on the gel, and correct the labeling of the polymerase fragments in the legends.

Reviewer #2: (1) (a) Add, in the Methods section, the relevant experimental details for VHH16 escape mutants generation and the subsequent characterization, presumably by deep sequencing. Parameters for variant calling should be mentioned.

(b) Why did the authors not validating S405N in their functional validation?

(2) Briefly discuss the proposed mechanism of action of Nb8191, and explain how it is compared with VHH16.

Reviewer #3: (No Response)

PLOS authors have the option to publish the peer review history of their article (what does this mean?). If published, this will include your full peer review and any attached files.

Reviewer #1: No

Reviewer #2: No

Reviewer #3: No

Figure Files:

Data Requirements:

Reproducibility:

References:

---

## [Editor Report · Decision Letter 2]

15 May 2024

Dear Dr Delmas,

We are pleased to inform you that your manuscript 'Antiviral activity of intracellular nanobodies targeting the influenza virus RNA-polymerase core' has been provisionally accepted for publication in PLOS Pathogens.

Best regards,

George A. Belov, PhD

Academic Editor

PLOS Pathogens

Meike Dittmann

Section Editor

PLOS Pathogens

Michael Malim

Editor-in-Chief

PLOS Pathogens

orcid.org/0000-0002-7699-2064
---

## [Editor Report · Acceptance letter]

11 Jun 2024

Dear Dr Delmas,

We are delighted to inform you that your manuscript, "Antiviral activity of intracellular nanobodies targeting the influenza virus RNA-polymerase core," has been formally accepted for publication in PLOS Pathogens.

Best regards,

Michael Malim

Editor-in-Chief

PLOS Pathogens

orcid.org/0000-0002-7699-2064